# MiR-155-5p Elevated by Ochratoxin A Induces Intestinal Fibrosis and Epithelial-to-Mesenchymal Transition through TGF-β Regulated Signaling Pathway In Vitro and In Vivo

**DOI:** 10.3390/toxins15070473

**Published:** 2023-07-22

**Authors:** Kyu Hyun Rhee, Seon Ah Yang, Min Cheol Pyo, Jae-Min Lim, Kwang-Won Lee

**Affiliations:** Department of Biotechnology, College of Life Sciences and Biotechnology, Korea University, Seoul 02841, Republic of Korea; rheekh02@korea.ac.kr (K.H.R.); y_suna@naver.com (S.A.Y.); reve03@naver.com (M.C.P.); jaemin1995@naver.com (J.-M.L.)

**Keywords:** ochratoxin A, miR-155-5p, intestinal fibrosis, TGF-β, EMT

## Abstract

Ochratoxin A (OTA) is a mycotoxin that induces fibrosis and epithelial-to-mesenchymal transitions (EMT) in kidneys and livers. It enters our bodies through food consumption, where it is absorbed in the intestines. However, the impact of OTA on the intestines is yet to be studied. MicroRNA (miRNAs) are small non-coding single-stranded RNAs that block the transcription of specific mRNAs and are, therefore, involved in many biochemical processes. Our findings indicate that OTA can induce EMT and intestinal fibrosis both in vivo and in vitro. This study examines the impact of OTA on intestinal toxicity and the role of miRNAs in this process. Following OTA treatment, miR-155-5p was the most elevated miRNA by next-generation sequencing. Our research showed that OTA increased miR-155-5p levels through transforming growth factor β (TGF-β), leading to the development of intestinal fibrosis and EMT. Additionally, the study identified that the modulation of TGF-β and miR-155-5p by OTA is linked to the inhibition of CCAAT/enhancer-binding protein β (C/EBPβ) and Smad2/3 accumulation in the progression of intestinal fibrosis.

## 1. Introduction

Ochratoxin A (OTA) is a mycotoxin produced by *Aspergillus* spp. and *Penicillium* spp. found abundantly in most cereals and their products, fruits and vegetables, meat, and poultry [1,2,3,4]. OTA has been categorized as a Group 2B carcinogen for humans according to the classification by the International Agency for Research on Cancer (IARC) in 1993. It is found in loaf, beer, dehydrated fruits, chicken, and wine, among other things [5]. Due to its abundance in food, OTA is easily introduced into the human body, where it accumulates in tissues and organs. OTA mostly affects the kidney, where it causes epithelial-to-mesenchymal transitions (EMT) and renal fibrosis [6]. Our previous study also reported that OTA also induced hepatic fibrosis and aryl hydrocarbon receptor (AhR)-regulated hepatotoxicity in liver tissues of mice [7,8]. However, there has been little study on the potentially toxic impact of OTA on the human gut. Given that humans consume OTA through food, it is essential to understand how OTA affects the intestines.

Intestinal fibrosis is usually considered to be a common complication of pathophysiology, such as inflammatory bowel disease (IBD), collagenous colitis, cystic fibrosis, and gastrointestinal stromal tumors (GISTs) leading to intestinal stenosis and obstruction [9]. One of the common signaling pathways responsible for fibrosis development is the transforming growth factor β (TGF-β)/Smad pathway, resulting in the Smad2/3 nucleus accumulation [6]. Intestinal fibrosis progression, like fibrosis in other organs, is directly or indirectly promoted by production of extracellular matrix by mesenchymal cells. The promotion of mesenchymal cells through EMT progression is therefore one of the key factors of fibrotic development; thus, it is common to relate EMT to fibrosis development [10]. The development of EMT and intestinal fibrosis triggered by OTA yet remains unknown.

MicroRNAs (miRNAs) are non-coding and single-stranded RNA molecules that are usually composed of 18–25 nucleotides [11,12]. MiRNAs target specific mRNAs to suppress protein synthesis post-transcriptionally [11]. Furthermore, miRNAs have been observed to activate gene expression in certain conditions [13]. The synthesis of miRNAs and their actions are crucial to cellular metabolisms and biochemical activities [14]. Unsurprisingly, miRNAs are known to play crucial roles in fibrosis development. miR-29 downregulation is known to be associated with fibrosis in heart, liver, kidney, and skin [15,16]. Lewis et al. [17] reviewed that miR-29 and miR-200 families are associated with early intestinal fibrosis progression. MiR-21, miR-133a, miR-192, and miR-17-5p are known as regulating miRNAs of TGF-β1-associated fibrotic development in various organs [18].

Among the myriad of miRNAs, miR-155-5p has been recognized to undergo upregulation in a TGF-β-enriched microenvironment. By targeting and inhibiting the expression of CCAAT/enhancer-binding protein beta (C/EBPβ), it promotes epithelial-mesenchymal transition (EMT) and metastasis in breast cancer [19]. Upregulation of miR-155-5p is also reported to induce renal fibrosis by targeting suppressor of cytokine signaling 1 (SOCS1) and 6 (SOCS6) [20]. We also reported that inhibition of miR-155-5p successfully downregulated EMT and fibrosis markers promoted by OTA, such as fibronectin (FN) and E-cadherin (E-cad) in kidney HK-2 cells [21]. However, the mechanism on whether miR-155-5p and TGF-β regulation affects the development of intestinal fibrosis is clouded.

This work intended to determine the incidence of EMT and fibrosis in mice colon tissues and human colon cells, as well as the underlying mechanisms involving miRNA activities and the TGF-β/Smad2/3 signaling pathway. We conducted a study on the changes in miRNA expression in the colon following OTA exposure, as well as the development of intestinal fibrosis influenced by 1 and 3 mg/kg B.W. of OTA in mice colon tissues, as well as 25 nM and 100 nM of OTA on Caco-2 cells.

## 2. Results

### 2.1. Histopathological Changes in Mice Colon Tissues after OTA Exposure

To examine the damage induction on colon tissue by OTA, ICR mice were orally administered OTA with low and high concentrations. The staining was performed using hematoxylin and eosin (H&E) and Masson’s trichrome (MT). H&E staining showed that OTA caused inflammatory infiltration in the submucosa (black arrow) and muscle layer (yellow arrow) dependent on its concentration (Figure 1a). MT staining demonstrated that OTA induced fibrosis (black arrow) in colon tissues as well (Figure 1b). The group administered with OTA at a high concentration (OH) exhibited a significant (*p* < 0.05) increase in areas stained with MT (Figure 1c). These results show that OTA caused intestinal damage and triggered fibrotic progression in mice colon tissues.

### 2.2. OTA Induced EMT and Fibrosis in Colon In Vivo and In Vitro

MT staining in mice colon tissues indicated the progression of fibrosis. Using the colon tissues and Caco-2 cells, we conducted in vivo and in vitro tests to measure the levels of E-cad, α-smooth muscle actin (α-SMA), FN, TGF-β, and TGF-β receptor 1 (TGF-β R1) for assessing the development of EMT as well as the production of fibrosis. In mice colon tissues, the levels of α-SMA, FN, TGF-β, and TGF-β receptor R1 (TGF-β R1) have shown to be upregulated whilst the level of E-cad has been downregulated under OTA exposure (Figure 2a,b). The OH group that received a high dosage of OTA had significantly (*p* < 0.05) lower E-cad protein levels than the control group. In regard to both FN and TGF-β R1, the expression levels of these proteins in the OH-treated group were substantially (*p* < 0.05) greater than the control group. Similarly, OTA exposure on Caco-2 cells also elevated the mRNA and protein levels of α-SMA, FN, TGF-β, and TGF-β R1 and simultaneously inhibited the mRNA and protein level of E-cad (Figure 3a,b). These results shows that OTA exposure triggers EMT and fibrosis progression in the colon.

### 2.3. Differently Expressed miRNAs in Mice Colon Tissues after OTA Exposure

miRNA expression changes were analyzed and identified by miR-seq. The colon tissues from 9 mice (*n* = 3 for each group) were subjected to next generation sequencing (NGS) analysis of miRNAs. Differently expressed miRNAs (DEMs) were showed and illustrated in Figure 4a,b. The comparison was made pairwise among control, OL, and OH groups. OTA exposure has induced 42 DEMs in the colon tissues of mice (*p* < 0.05 and fold change > 1.5).

The miRNA showing the highest upregulation was miR-155-5p (Figure 4b). This observation was corroborated in mouse colon tissues and Caco-2 cells, where miR-155-5p exhibited a significant (*p* < 0.05) increase in expression upon exposure to OTA (Figure 4c,d). This led us to investigate the relationship between EMT and fibrotic development in the colon and miR-155-5p regulation.

### 2.4. miR-155-5p Targets C/EBPβ under OTA Treatment

To find out if miR-155-5p was engaged in EMT and fibrotic development in the colon, we searched the TargetScan database for miR-155-5p target mRNAs. (https://www.targetscan.org/vert_80/, accessed on 17 February 2023). Among the several putative target mRNAs of miR-155-5p, C/EBPβ has been linked to TGF-regulated fibrotic progression [22]. Except for the C/EBPβ mRNA expression level in the OH group, the mRNA and protein levels of C/EBPβ were both significantly (*p* < 0.05) downregulated under OTA treatment in mice colon tissues and Caco-2 cells (Figure 5). C/EBPβ is a known target gene of miR-155-5p [19], leading us to postulate that miR-155-5p is connected to the decrease of C/EBPβ under the influence of OTA.

### 2.5. Regulation of EMT and Fibrosis by miR-155-5p under OTA Treatment

To further unveil the mechanism on which miR-155-5p is regulated in fibrotic progression, specifically with the TGF-β regulated pathway, we used GW788388, a TGF-β specific inhibitor, and checked for EMT and fibrotic marker expressions. Treatment with GW788388 for 1 h, followed by 100 nM of OTA for 48 h, significantly (*p* < 0.05) suppressed the expression of miR-155-5p (Figure 6a). TGF-β inhibition significantly (*p* < 0.05) reduced α-SMA and FN expression while restoring C/EBP and E-cad expression (Figure 6b,c). Thus, the upregulation of TGF-β after OTA treatment is responsible for miR-155-5p upregulation and the subsequent progression of EMT and fibrosis.

We also used miR-155-5p inhibitor to determine the role of miR-155-5p in this fibrotic progression, and it successfully knocked down miR-155-5p expression (Figure 7a). Inhibition of miR-155-5p restored the expressions of C/EBPβ and E-cad, whilst it downregulated α-SMA and FN expressions (Figure 7b,c). Interestingly, the mRNA level of TGF-β and protein level of TGF-β R1 remained upregulated after inhibition of miR-155-5p under OTA treatment (Figure 7b,c). This demonstrates that TGF-β expression is unaffected by miR-155-5p and that the expression level of miR-155-5p is only increased when TGF-β is present during OTA treatment.

The above results show that TGF-β upregulates miR-155-5p expression, leading to downregulation of C/EBPβ, which ultimately leads to EMT and fibrosis progression.

### 2.6. miR-155-5p Induces Fibrosis under OTA Treatment by Regulating TGF-β/Smad/2/3 Pathway

In our previous study, we showed that OTA had induced renal fibrosis through the TGF-β/Smad/2/3 signaling pathway [6]. In accordance with the study, fractionation of cells and tissues were performed to determine whether OTA induced nuclear accumulation of Smad2/3. As shown in Figure 8a,b, it was similarly observed in vivo and in vitro. In addition, inhibition of miR-155-5p had slightly decreased the nuclear accumulation of Smad2/3 (Figure 8c), which indicates miR-155-5p elevation along with C/EBPβ inhibition plays a role in Smad2/3 nuclear accumulation. This suggests that OTA causes intestinal fibrosis via TGF-β/Smad2/3 signaling pathway with miR-155-5p regulation.

## 3. Discussion

OTA is a mycotoxin found in food that is known to be nephrotoxic, immunotoxic, carcinogenic, and teratogenic to animal species [23]. When consumed excessively, OTA causes different types of damage throughout the body, mainly in the kidney [24]. However, OTA affects not only the kidneys but also the liver and colon. OTA induced hepatotoxicity through different pathways including sulfur amino acid metabolism (cysteine and methionine), hepatic bile acid biosynthesis, and xenobiotic biotransformation by cytochrome P450 [25]. OTA is known to induce liver inflammation through the regulation of intestinal microbiota, while maintenance of intestinal microbiota homeostasis alleviates the effect [26]. We also reported that OTA, along with co-treatment of cadmium, had caused severe intestinal dysfunction in the intestines [27].

It should be emphasized that the doses of OTA used in the current animal study, as well as in numerous previous studies referenced in the literature [28,29], were orders of magnitude higher than the average exposure of the human population to OTA [30]. The dosage regimen for this study was determined in accordance with the findings from recent murine investigations assessed by the European Food Safety Authority [30]. The selection of the lower dose (1 mg/kg body weight (B.W.)) was based on the lowest observed adverse effect level (LOAEL) identified in a 45-day oral study, where a dose of 1.5 mg/kg B.W. elicited a response in kidney antioxidants. Our experiments also included a higher dose (3 mg/kg B.W.). Our in vitro experiments utilized OTA concentrations as low as 25 nM, which were determined based on previous research. Renal and liver toxicity was documented by Pyo M.C. et al. [31] and Shin H.S. et al. [8] at an OTA concentration of 200 nM. Additionally, Gao Y. et al. [32] employed concentrations of 0.2 μM and 20 μM in their study involving OTA treatment on Caco-2 cells. However, it is important to note that that our selected concentrations only represent levels associated with high OTA intake leading to intoxications [6,28,29], and they are approximately 1000 times lower than the blood concentration observed in healthy individuals. A study involving 2000 Swedish adults reported a maximum blood concentration of OTA at 0.136 μg/L [33].

In this study, we attempted to uncover fibrosis development in the intestines when OTA was consumed excessively through contaminated food. Similar to our previous study with the effect of OTA on the kidney and liver [6,8], OTA is found to induce fibrosis in mouse colon intestine as shown by MT staining in mice colon tissues. Our results also showed upregulation of FN and α-SMA, which are EMT and fibrosis markers, along with downregulation of epithelial phenotype marker E-cad. OTA also elevated both TGF-β and Smad2/3 expressions, along with accumulation of Smad2/3 in the nucleus region of cells, indicating the involvement of the TGF-β/Smad2/3 signaling pathway is the fibrotic progression. The TGF-β/Smad2/3 pathway is one of the mainstream fibrosis processing mechanisms, triggered by TGF-β expression and resulting in the phosphorylation of Smad2 and Smad3 and forming a complex via Smad signaling [34].

MiRNAs bind to the 3′ UTR of target mRNAs and decrease their expression, giving miRNAs an important role in different signaling pathways and cellular metabolism [35]. Through NGS analysis of DEMs in our study, miR-155-5p was the highest increased miRNA across all DEMs. As a result, we approached our investigation of intestinal fibrosis and EMT through the control of miR-155-5p. In a prior study, after OTA treatment, the levels of miR-155-5p, along with FN and α-SMA, were significantly enhanced [21]. Inhibiting miR-155-5p also delayed the formation of fibrosis in HK-2 kidney cells. It also caused pulmonary fibrosis by targeting FOXO3a, a TGF-β-regulating gene [36], and by activating the NLRP3 inflammasome in bleomycin-induced pulmonary fibrotic rat lungs [37]. Furthermore, lipopolysaccharide-induced miR-155-5p elevation is recognized to be a cause of liver fibrosis via alteration of gut permeability led by inflammatory and fibrotic response in the intestine [18]. These suggest that increased miR-155-5p expression leads to fibrosis development in the kidney, lung, and liver. In our study, we observed an increase in miR-155-5p expression in both mice colon tissues and Ca-co-2 cells when exposed to OTA. Interestingly, the inhibition of miR-155-5p resulted in decreased expressions of FN and α-SMA, while the expression of E-cad was increased. These findings lead us to infer that miR-155-5p is likely involved in the progression of fibrosis in the intestine as well. A comprehensive assessment of the signaling pathway underlying the interaction between miR-155-5p and fibrosis development is required.

MiR-155-5p has been identified to target C/EBPβ, and inhibiting C/EBPβ through fmiR-155-5p can lead to EMT, invasion, and metastasis in breast cancer [19]. C/EBPβ is known to be a factor in the TGF-β-regulated signaling pathway, including fibrotic progressions [38]. In accordance with the study in breast cancer, our studies revealed that C/EBPβ had decreased under OTA treatment, while the expression of miR-155-5p had increased. C/EBPβ involvement in fibrotic development is somewhat divisive, since both elevation and downregulation of C/EBPβ protein levels have been observed in hepatic and pulmonary fibrosis [39]. In our investigation, however, miR-155-5p inhibition restored C/EBP expression, suggesting that OTA treatment promoted miR-155-5p expression, leading to C/EBPβ loss. Furthermore, TGF-β inhibition increased C/EBPβ expression, which was blocked by miR-155-5p expression. Interestingly, the inhibition of miR-155-5p did not affect TGF-β expression, suggesting that TGF-β might be responsible for activating miR-155-5p. These findings suggest that OTA treatment of the colonic intestine induced TGF-β expression, which was then followed by an increase in miR-155-5p and a reduction of C/EBPβ. Nevertheless, further studies should uncover and validate the exact mechanism of the regulation of fibrotic progression by miR-155-5p and C/EBPβ in TGF-β mediated pathway.

TGF-β is a critical regulator of fibrotic development in the kidney, liver, lung, and heart by phosphorylating Smad2 and Smad3 proteins [40]. Similarly, Yun et al. [41] discovered that TGF-β mediates intestinal fibrosis. It is uncertain if TGF-β has a similar function to fibrosis in other organs in the development of intestinal fibrosis. The nuclear and cytoplasmic fractionization in this study showed that Smad2 and Smad3 proteins were accumulated in the nuclear region, like our previous study with the kidney [6]. Although additional research is required to fully understand Smad2 and Smad3 involvement in intestinal fibrosis, it is reasonably inferred that TGF-β is involved in intestinal fibrosis development, as well as in other organs. Furthermore, miR-155-5p suppression reduced fibrotic development in Caco-2 cells by decreasing Smad2 and Smad3 protein accumulation in the nucleus area of the cell. Therefore, it can be anticipated that the regulations of TGF-β and miR-155-5p play a crucial role in the nuclear accumulation of Smad2 and Smad3 during the development of intestinal fibrosis.

In summary, our research indicates that OTA induces intestinal EMT-associated fibrosis by modulating the regulation of TGF-β and miR-155-5p. Although the precise relationship between C/EBPβ and Smad2/3 requires further investigation, the inhibition of C/EBPβ by miR-155-5p and the nuclear accumulation of Smad2 and Smad3 may play significant roles in the development of OTA-induced intestinal fibrosis. These findings may contribute to our understanding of the involvement of miR-155-5p in intestinal fibrosis and suggest its potential as a therapeutic target for addressing intestinal fibrosis and related conditions. Nevertheless, further studies are necessary to completely understand the function and precise interaction of TGF-β, miR-155-5p, and C/EBPβ.

## 4. Materials and Methods

### 4.1. Materials

Cfm Oskar Tropitzsch GmbH (Marktredwitz, Germany) supplied the OTA. Hyclone (Logan, UT, USA) provided fetal bovine serum (FBS), antibiotics (penicillin–streptomycin), and trypsin-ethylenediaminetetraacetic acid. Dulbecco’s Modified Eagle Medium (DMEM) was obtained from Gibco (Grand Island, NY, USA). miR-155-5p inhibitor (339121) and miRNA inhibitor control (339126) were obtained from Qiagen (Valencia, CA, USA). GW788388, a TGF-β R1 inhibitor, was purchased from Sigma-Aldrich (St. Louis, MO, USA). Primary antibodies against FN (sc-8422), α-SMA (sc-53142), TGF-β R1 (sc-101574), C/EBPβ (sc-7962), Smad2/3 (sc-133098), and GAPDH (sc-32233) were purchased from Santa Cruz Biotechnology (Dallas, TX, USA). Cell Signaling (Denvers, MA, USA) supplied E-cad (14472S).

### 4.2. Animal Study

The animal study was carried out with the approval and compliance of Korea University’s Committee for Ethical Use of Experimental Animals (KUIACUC-2021-0026). The design and procedure of the animal study conducted in this paper were described in detail in the previous report [21]. The mice were divided into 3 groups (*n* = 6 per groups): (1) CON group: 0 mg/kg B.W. of OTA, (2) OL group: 1 mg/kg B.W. of OTA, and (3) OH group: 3 mg/kg B.W. of OTA. After OTA treatment for 12 weeks, the colon tissues of 6 mice per group were obtained for subsequent study.

### 4.3. Histopathological Analysis

Mouse colon tissues were preserved in 10% formalin solution. Subsequently, they were fixed in paraffin and sectioned into slices of 4–6 μm thickness using a rotating microtome (Leica Microsystems Ltd., Melbourne, Australia). Then, they were deparaffinized and stained with H&E and MT before being visualized under an optical microscope (Olympus, Tokyo, Japan) and a digital slide scanner (VM1, Motic, Beijing, China). Quantification of the collagen deposition was performed using the Solution for Automatic Bio-Image Analysis (SABIA) software (EBIOGEN, Seoul, Republic of Korea). Three mice colon tissues were chosen randomly from each group for this study (*n* = 3).

### 4.4. microRNA-Sequencing (miR-seq)

Three mice colon tissues from each group were randomly selected and subjected to miRNA-sequencing. Sequence reads were mapped and read counts were identified as previously described [21]. The normalization method of counts per million and trimmed mean of M-values (CPM + TMM) was also used. The differential expression markers (DEMs) were identified based on the criterion suggested by Béres N.J. et al. [42] and Nassirpour R. et al. [43], where a fold change greater than 1.5 was used as the threshold.

### 4.5. Cell Culture and Treatment

Human intestinal Caco-2 cells (ATCC HTB-37) were purchased from American Type Culture Collection (Rockville, MD, USA). Caco-2 cells were maintained in Dulbecco’s modified Eagle’s medium (low glucose, Gibco, Grand Island, NY, USA) supplemented with 3.7 g/L sodium bicarbonate, 1% (*v*/*v*) penicillin/streptomycin, and 10% (*v*/*v*) fetal bovine serum (FBS). The cells were then cultured in a controlled environment within a humidified incubator at constant conditions (5% CO_2_ and 37 °C). Before OTA treatment, cells were seeded at 6-well plates for 5 × 10^5^ cells/well. OTA was treated by dissolving it in dimethyl sulfoxide (DMSO) and diluting it with DMEM-low glucose media to final concentrations of 25 nM and 100 nM, where the final concentration of DMSO was 0.1%.

### 4.6. Cell Transfection

To inhibit miR-155-5p expression, Caco-2 cells were transfected with miR-155-5p inhibitor using Lipofectamine^TM^ RNAiMAX transfection reagent (Invitrogen, Carlsbad, CA, USA) for 48 h according to the manufacturer’s protocol. The cells were seeded to 6-well plates with count of 2.5 × 10^5^ cells/well.

### 4.7. Isolation of Nuclear and Cytosolic Extracts

Nuclear and cytosolic fractions of Caco-2 cells were obtained following the previously established protocols with comprehensive descriptions [44]. Nuclear and cytosolic fractions of mouse colon tissue were obtained through a tissue fractionation protocol [45]. The extracts containing nuclear and cytosolic protein were stored at −70 °C.

### 4.8. Quantitative Real-Time PCR (qRT-PCR) Analysis

Total RNA of colon tissues and Caco-2 cells were extracted using RNAiso Plus (Takara, Kusatu, Japan). cDNA was synthesized using first-strand cDNA synthesis kit (LeGene Biosciences, San Diego, CA, USA). miRNAs were isolated using miRNeasy Tissue/Cells Advanced Mini Kit (Qiagen, Valencia, CA, USA). The mRNA expressions were evaluated using EzAMP^TM^ Real-Time qPCR 2X Master Mix SYBR Green (ELPIS Biotech, Daejeon, Korea) and were normalized by GAPDH expression. Similarly, the miRNA expressions were measured using miRCURY LNA SYBR Green PCR kit (Qiagen, Valencia, CA, USA) and were normalized by U6 expression. The primer sequences of mRNAs used in this study are presented in the Appendix A.

### 4.9. Western Blot Analysis

RIPA buffer (EBA-1149, Elpis Biotech Inc., Seoul, Republic of Korea) containing 5 μg/mL of aprotinin and leupeptin were used to extract proteins from colon tissues and Caco-2 cells. Protein lysates were obtained as supernatant collection after centrifugation for 20 min at 4 °C, 13,000 rpm. Lysates (40 μg) was separated by 8–14% gel electrophoresis and were transferred to polyvinylidene difluoride membranes (Millipore, Billerica, MA, USA). Membranes were subjected to blocking in 5% skim milk in TBST (Tris-buffered saline with 0.1% Tween 20) before being incubated overnight with primary antibodies overnight against FN (1:500), α-SMA (1:500), E-cadherin (1:1000), TGF-β R1 (1:500), C/EBPβ (1:500), Smad2/3 (1:500), and GAPDH (1:1000) at 4 °C. The membranes were then incubated for 1 h with a goat anti-mouse horseradish peroxidase-conjugated secondary antibody (Millipore, Billerica, MA, USA) before the protein bands were identified using an ImageQuantTM LAS 400 mini (GE Healthcare, Buckinghamshire, UK). The protein bands were visualized by using ECL™ Select Western blotting detection reagent (Cytiva, Buckinghamshire, UK). ImageJ analysis (National Institutes of Health, Bethesda, MD, USA) was used to quantify the bands. As an internal control, GAPDH was employed.

### 4.10. Statistical Analysis

All data are expressed as the mean ± standard deviation (SD). Statistical analysis was conducted using a one-way analysis of variance (ANOVA) followed by Duncan’s multiple range test for in vivo experiments and Tukey’s multiple range test for in vitro studies. Different letters above the bars represent significant differences at *p*-value lower than 0.05 using SAS version 9.4 (SAS Institute, Cary, NC, USA).

## Figures and Tables

**Figure 1 toxins-15-00473-f001:**
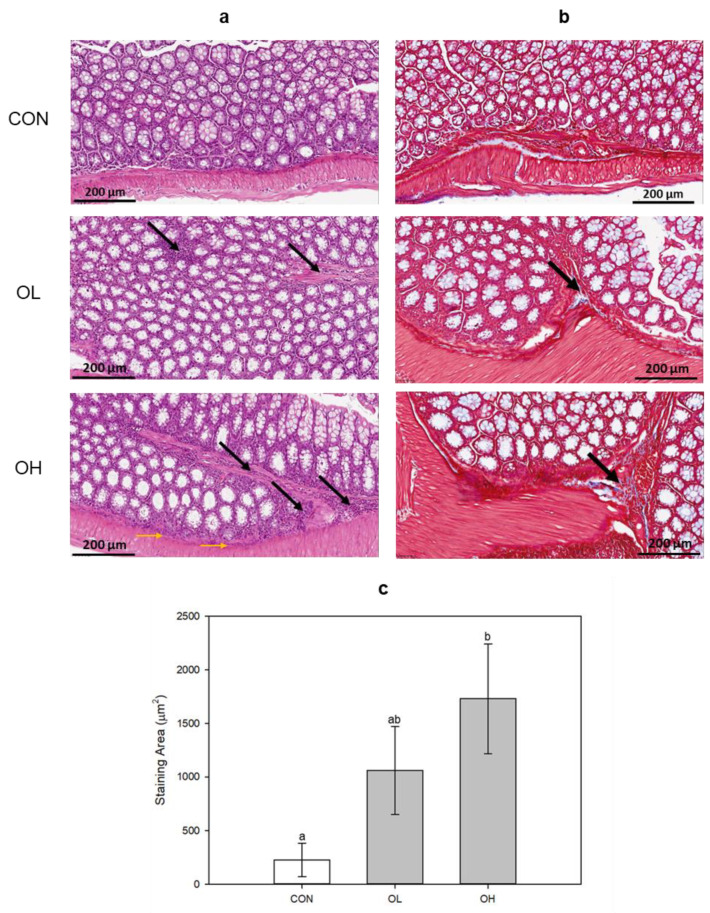
Histopathological changes in colon tissues by (**a**) H&E and (**b**) MT staining. (**c**) Areas stained with MT were expressed (μm^2^). In H&E (original magnification × 40; scale bar, 200 μm), the black arrows indicate submucosa inflammatory infiltration (neutrophil and lymphocyte), and the yellow arrows indicate muscle layer inflammatory infiltration. In MT staining (original magnification × 40; scale bar, 200 μm), the black arrows indicate the location of fibrosis. CON: untreated control; OL: ochratoxin A (OTA)-low (1 mg OTA/kg body weight (B.W.)); OH: OTA-high (3 mg OTA/kg B.W.). Data represent the mean ± S.D. of experiments with 3 mouse colon tissues per group, and significance was determined compared with the control by Duncan’s studentized range test; different letters mean significant differences at *p* < 0.05.

**Figure 2 toxins-15-00473-f002:**
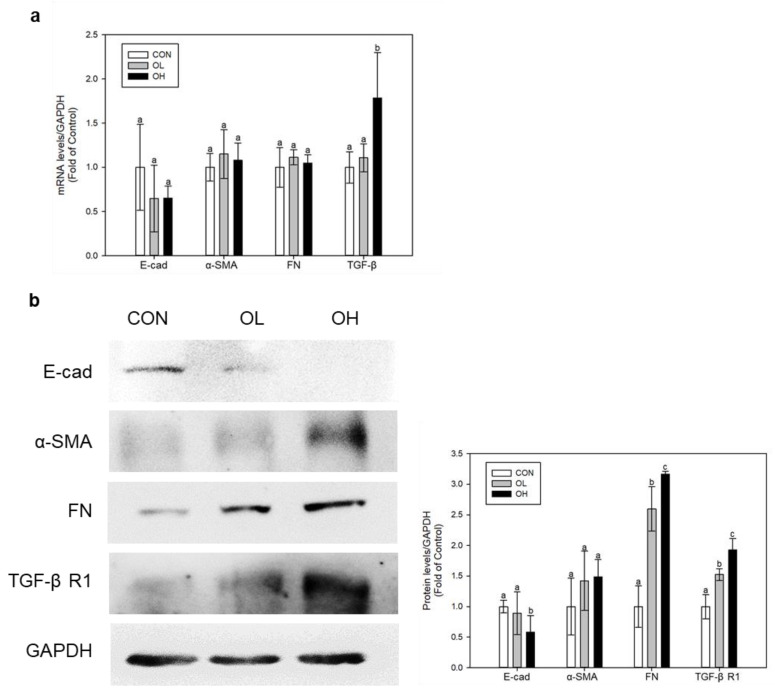
OTA induces EMT and fibrosis in mouse colon tissues. (**a**) mRNA and (**b**) protein expressions were determined using real-time qPCR and Western blot, respectively. Data represent the mean ± S.D. of experiments with 6 mouse colon tissues per group, and significance was determined by comparing with the control by Duncan’s studentized range test. Different letters indicate significant differences at *p* < 0.05.

**Figure 3 toxins-15-00473-f003:**
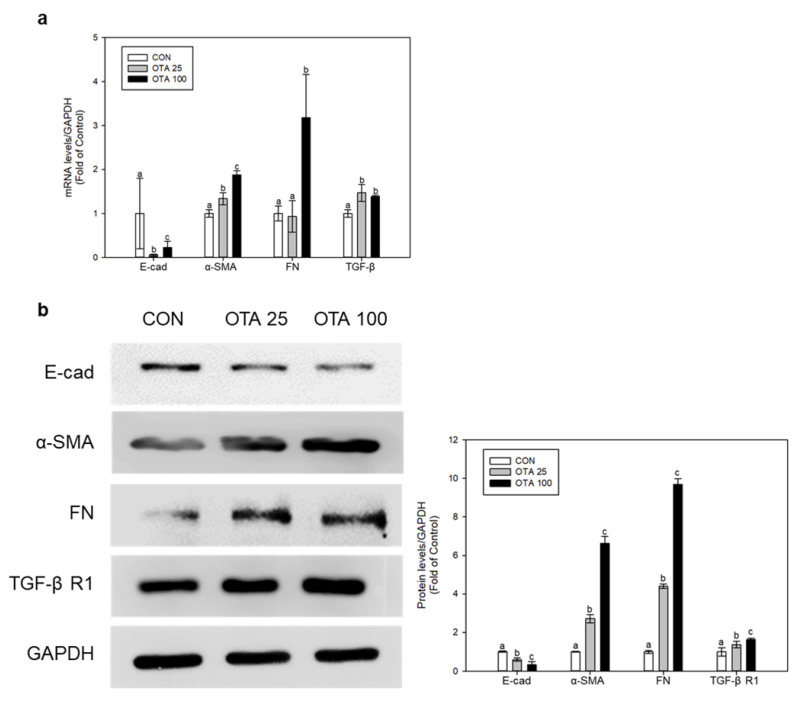
OTA induces EMT and fibrosis in human colon cell line Caco-2. Caco-2 cells received treatment with 25 and 100 nM of OTA for 48 h. (**a**) mRNA and (**b**) protein expressions were determined using real-time qPCR and Western blot, respectively. All mRNA and protein levels were normalized by GADPH expression. Data represent the mean ± S.D. of experiments with triplicate samples, and significance was determined by comparing with the control using Tukey’s studentized range test and different letters mean significant differences at *p* < 0.05. CON: untreated control; OTA 25: 25 nM OTA; OTA 100: 100 nM OTA.

**Figure 4 toxins-15-00473-f004:**
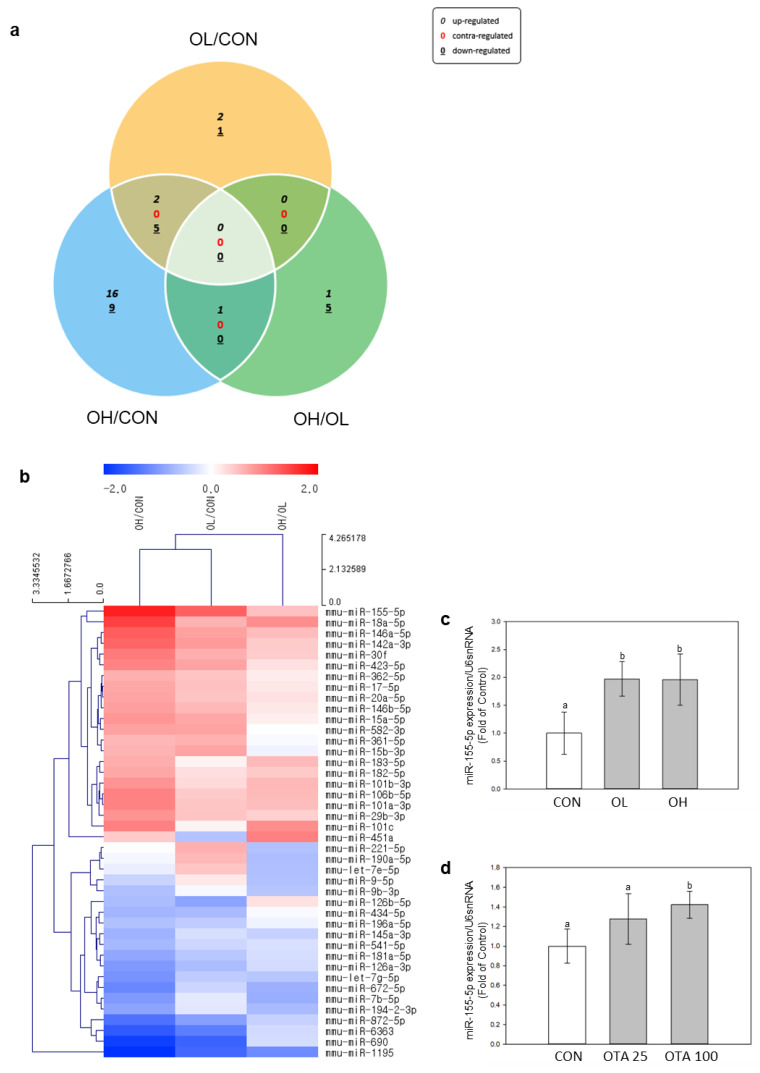
Identification of transcriptomes by next generation sequencing (NGS) in mice colon tissues exposed to OTA. Expression patterns of DEMs were expressed in (**a**) Venn diagrams and (**b**) heatmaps. Venn diagrams indicated overlapping DEMs by pairwise comparisons OL/CON, OH/CON, and OL/OH. DEM heatmaps were displayed as red for upregulated miRNAs and blue for downregulated miRNAs of pairwise comparison between OL/CON, OH/CON, and OL/OH. The DEMs were aligned to decreasing order of fold change. All DEMs in the heatmap analysis showed a 1.5-fold change in cut-off values, *p* < 0.05. Validation of miRNA NGS analysis of miR-155-5p in (**c**) mice colon tissues and (**d**) Caco-2 cells were performed by measuring miR-155-5p expressions using real-time qPCR. The expressions of miR-155-5p were normalized by U6 snRNA expression. Data represent the mean ± S.D. of experiments with triplicate samples, and significance was determined by comparing with the control using Tukey’s studentized range test; different letters mean significant differences at *p* < 0.05.

**Figure 5 toxins-15-00473-f005:**
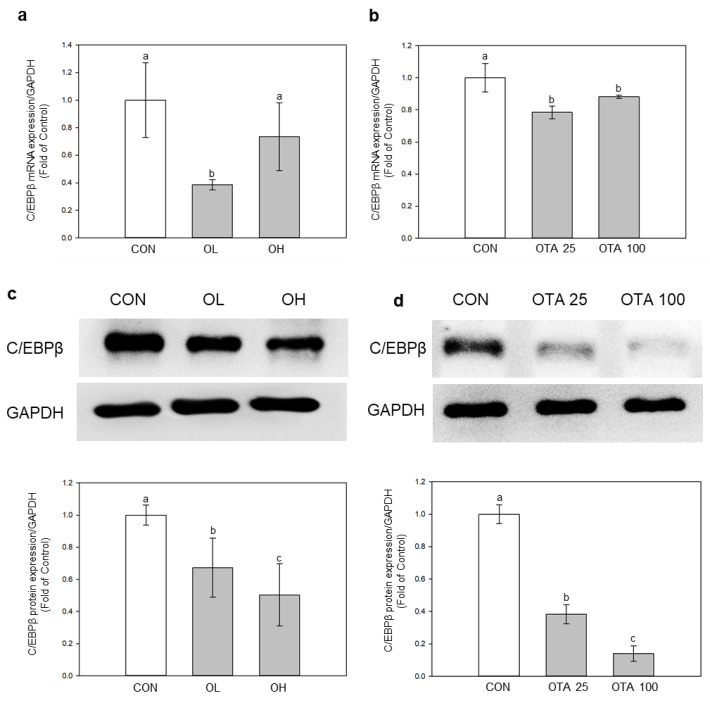
Identification of C/EBPβ as miR-155-5p target. mRNA expression of C/EBPβ (**a**) in vivo and (**b**) in vitro was evaluated. Protein expression of C/EBPβ (**c**) in vivo and (**d**) in vitro was evaluated. Data represent the mean ± S.D. of experiments with 6 mouse colon tissues per group (in vivo) and 3 experiments with triplicate samples (in vitro), and significance was determined by comparing with the control using Duncan’s studentized range test (in vivo) and Tukey’s studentized range test (in vitro); different letters mean significant differences at *p* < 0.05.

**Figure 6 toxins-15-00473-f006:**
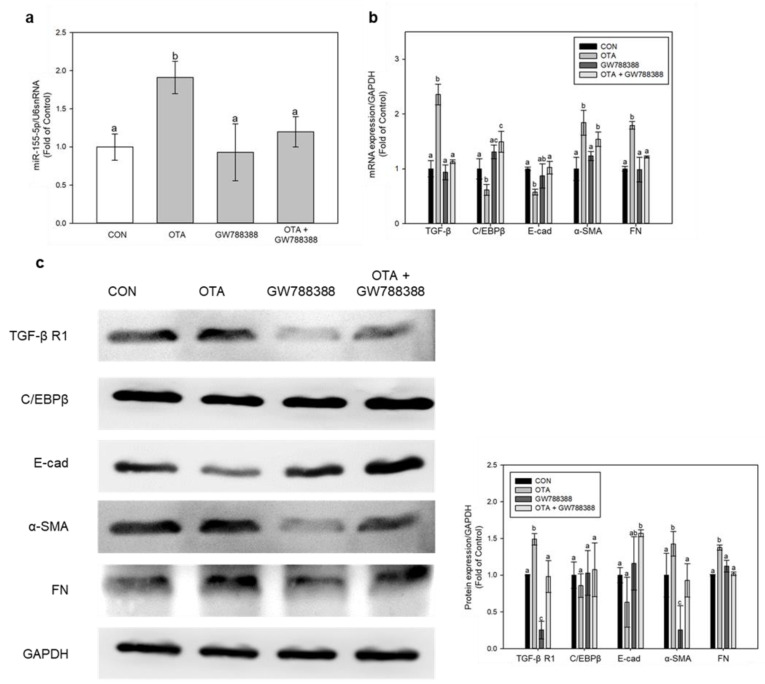
Regulation of intestinal fibrosis and EMT by TGF-β inhibition using GW788388. (**a**) The expressions of miR-155-5p were measured, and (**b**) mRNA and (**c**) protein expressions of fibrosis and EMT markers were examined. Data represent the mean ± S.D. of 3 experiments with triplicate samples, and different letters mean significant differences at the *p* < 0.05 by Tukey’s studentized range test.

**Figure 7 toxins-15-00473-f007:**
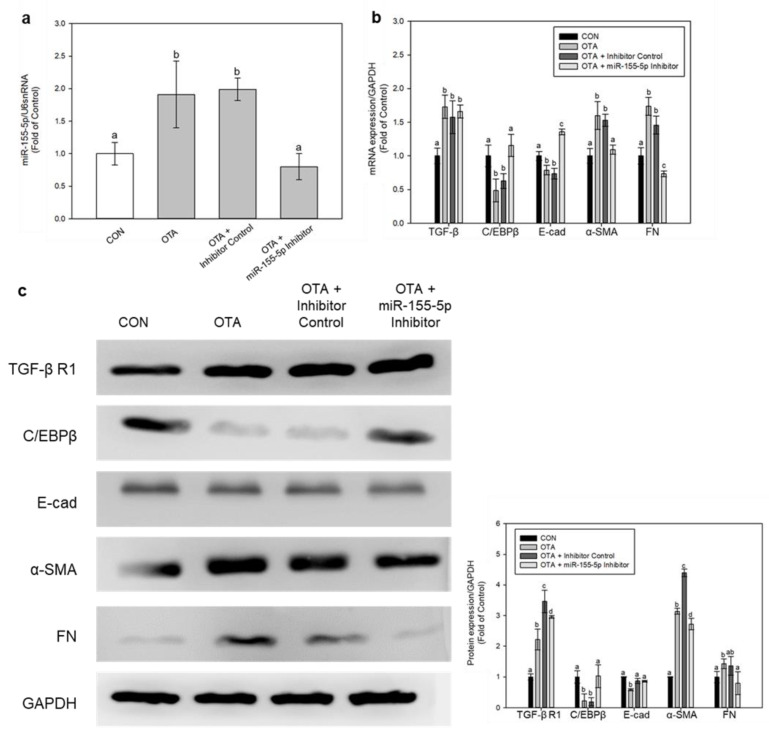
Regulation of intestinal fibrosis and EMT by miR-155-5p inhibition. (**a**) The expression of miR-155-5p was validated using a real-time PCR and normalized by U6 snRNA expression. Genes of fibrosis and EMT were evaluated by (**b**) real-time qPCR and (**c**) Western blotting. Data represent the mean ± S.D. of 3 experiments with triplicate samples, and different letters mean significant differences at the *p* < 0.05 by Tukey’s studentized range test.

**Figure 8 toxins-15-00473-f008:**
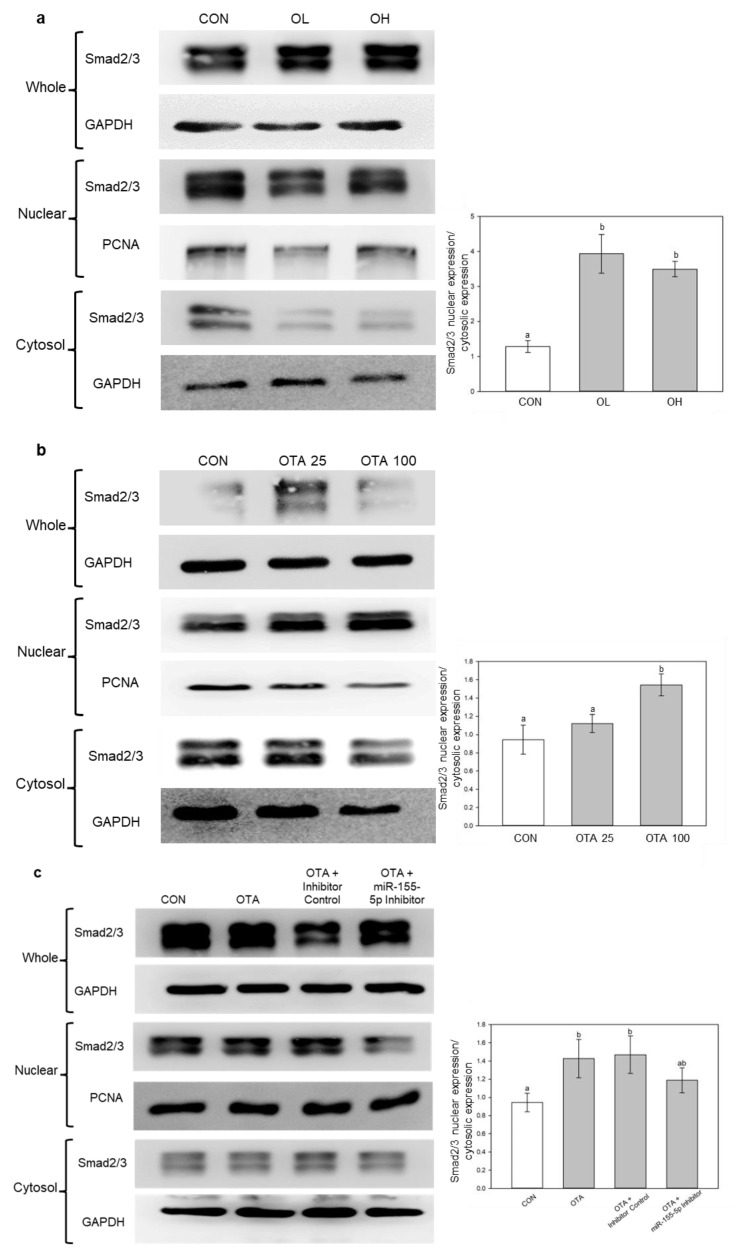
miR-155-5p regulates fibrosis by the activation of TGF-β/Smad2/3 pathway. Smad2/3 levels were analyzed by Western blotting (**a**) in vivo and (**b**) in vitro. The levels of Smad2/3 in cytosol and nuclear after miR-155-5p inhibition were analyzed in (**c**). Data represent the mean ± S.D. of experiments with 6 mouse colon tissues per group (in vivo) and 3 experiments with triplicate samples (in vitro), and significance was determined by comparing with the control using Duncan’s studentized range test (in vivo) and Tukey’s studentized range test (in vitro); different letters mean significant differences at *p* < 0.05.

## Data Availability

The data presented in this study are available in the article or Appendix A here.

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
