# Peer review of "MiR-155-5p Elevated by Ochratoxin A Induces Intestinal Fibrosis and Epithelial-to-Mesenchymal Transition through TGF-β Regulated Signaling Pathway In Vitro and In Vivo"

_toxins, 2023, doi:10.3390/toxins15070473_

Round 1

Reviewer 1 Report

Although the article has done a lot of work, there are still many problems, especially the statistical analysis of some data that does not match the bands. The article must undergo major revisions before being accepted for publication.

1. Abbreviations that first appear in the paper should be annotated, such as EMT in the abstract, α-SMA and TGF- β R1 in line 94. And check in full paper.

2. line 27: Regarding OTA contamination, the following references (Frontiers in Microbiology, 2018, 9: 1386; Toxin Reviews, 2017, 36(1): 39-44; Food and Chemical Toxicology, 2023, 176: 113793; Food and Chemical Toxicology, 2023, 172: 113592; Toxicology, 2021, 450: 152681) are very comprehensive and it is recommended to cite the references to supplement OTA-contaminated foods.

3. Fig. 2a TGF-β: Why is b missing? And check all figures.

4. Why were different measurement indicators selected in qPCR and Western blot experiments? One is the mRNA expression of TGF-β, another is the protein expression of TGF-β R1.

5. Fig. 3a: It is recommended to label statistical identifiers in alphabetical order. And check all figures.

6. Fig. 3b-c, Fig.6 c, and Fig.7 c: The statistical analysis of α-SMA does not match the result of the strip.

7. Fig. 4a: Why is there only OL/CON, but no OH/OL on Venn Diagram?

8. line 303: Why choose 1.5-fold as the cut-off value? References are needed.

9. The ordinate of Fig. 5 should use different titles. Other figures with the same issue should also be modified accordingly.

10. line 151: Due to the many targets of miR-155-5p, it does not seem to indicate that miR-155-5p targets C/EBP β by Fig. 4 and Fig. 5. At most, there is a connection between miR-155-5p and C/EBP β.

11. Fig.7 c: The statistical analysis of C/EBP β does not match the result of the strip.

12. The resolution of the ordinate of Fig. 8 is too low to allow for clear viewing.

Minor editing of English language required

Author Response

Reviewer 1

Although the article has done a lot of work, there are still many problems, especially the statistical analysis of some data that does not match the bands. The article must undergo major revisions before being accepted for publication.

  1. Abbreviations that first appear in the paper should be annotated, such as EMT in the abstract, α-SMA and TGF- β R1 in line 94. And check in full paper.

Þ Answer:

First of all, we’d like to express our gratitude to the reviewer for the careful and critical reading of our manuscript. With highlighted in green color text, we made the correction in response.

We have made the corrections in page (P) 1, line (L) 9 as follows:

“Ochratoxin A (OTA) is a mycotoxin that induces fibrosis and epithelial-to-mesenchymal transitions (EMT) in kidneys and livers.”

P4, L101-102 as follows:

“Using the colon tissues and Caco-2 cells, we conducted in vivo and in vitro tests to measure the levels of E-cad, α-smooth muscle actin (a-SMA), FN, TGF- β, and TGF- β receptor 1 (TGF- β R1) for assessing the development of EMT as well as the production of fibrosis.”

  1. line 27: Regarding OTA contamination, the following references (Frontiers in Microbiology, 2018, 9: 1386; Toxin Reviews, 2017, 36(1): 39-44; Food and Chemical Toxicology, 2023, 176: 113793; Food and Chemical Toxicology, 2023, 172: 113592; Toxicology, 2021, 450: 152681) are very comprehensive and it is recommended to cite the references to supplement OTA-contaminated foods.

Þ Answer:

We appreciate your suggestion and have cited the mentioned articles in P1, L29 and P15-16, L428-as follows:

“Ochratoxin A (OTA) is a mycotoxin produced by Aspergillus spp. and Penicillium spp. found abundantly in most cereals and their products, fruits and vegetables, meat, and poultry [1-4].”

  1. Cabanes, F.J.; Bragulat, M.R.; Castella, G. Ochratoxin A producing species in the genus Penicillium. Toxins (Basel) 2010, 2, 1111-1120, doi:10.3390/toxins2051111.
  2. Chen, W.; Li, C.; Zhang, B.; Zhou, Z.; Shen, Y.; Liao, X.; Yang, J.; Wang, Y.; Li, X.; Li, Y.; et al. Advances in Biodetoxification of Ochratoxin A-A Review of the Past Five Decades. Front Microbiol 2018, 9, 1386, doi:10.3389/fmicb.2018.01386.
  3. Deng, H.; Chen, W.; Zhang, B.; Zhang, Y.; Han, L.; Zhang, Q.; Yao, S.; Wang, H.; Shen, X.L. Excessive ER-phagy contributes to ochratoxin A-induced apoptosis. Food Chem Toxicol 2023, 176, 113793, doi:10.1016/j.fct.2023.113793.
  4. Zhang, Q.; Chen, W.; Zhang, B.; Li, C.; Zhang, X.; Wang, Q.; Wang, Y.; Zhou, Q.; Li, X.; Shen, X.L. Central role of TRAP1 in the ameliorative effect of oleanolic acid on the mitochondrial-mediated and endoplasmic reticulum stress-excitated apoptosis induced by ochratoxin A. Toxicology 2021, 450, 152681, doi:10.1016/j.tox.2021.152681.

  1. Fig. 2a TGF-β: Why is b missing? And check all figures.

Þ Answer:

We have made the change in Figure 2a. Please see Fig. 2a in P4.

  1. Why were different measurement indicators selected in qPCR and Western blot experiments? One is the mRNA expression of TGF-β, another is the protein expression of TGF-β R1.

Þ Answer:

Unfortunately, we were unable to conduct the ELISA assay to measure the protein expression of TGF-b due to the time constraint for the revision date. Please note that TGF-β is synthesized and secreted in a latent, biologically inactive form, which must be activated before binding to TGF-β receptors (TGF-β R1) (Chen A et al., 2002). In order to verify the activation of the TGF-β pathway, we conducted Western blot analysis specifically targeting TGF-β R1. This approach was taken based on the findings reported by Pyo MC et al. (2020) in our previous study.

Chen A, Davis BH, Sitrin MD, Brasitus TA, Bissonnette M. Transforming growth factor-β1 signaling contributes to Caco-2 cell growth inhibition induced by 1, 25 (OH) 2D3. American Journal of Physiology-Gastrointestinal and Liver Physiology. 2002 Oct 1;283(4):G864-74.

Pyo MC, Chae SA, Yoo HJ, Lee KW. Ochratoxin A induces epithelial-to-mesenchymal transition and renal fibrosis through TGF-β/Smad2/3 and Wnt1/β-catenin signaling pathways in vitro and in vivo. Archives of Toxicology. 2020 Sep;94:3329-42.

  1. Fig. 3a: It is recommended to label statistical identifiers in alphabetical order. And check all figures.

Þ Answer:

We have made the change Figure 3a in P4 and checked all figures.

  1. Fig. 3b, Fig.6c, and Fig.7c: The statistical analysis of α-SMA does not match the result of the strip.

Þ Answer:

We appreciate your precise readings. We have made the change in Figure 3b, 6c, and 7c.

  1. Fig. 4a: Why is there only OL/CON, but no OH/OL on Venn Diagram?

Þ Answer:

In fact, there were some overlapping labels (OH/OL) on the diagram, although they may not have been clearly visible. Consequently, we have increased the spacing between Fig. 4a and Fig. 4b to improve visibility. Please see Figure 4.

  1. line 303: Why choose 1.5-fold as the cut-off value? References are needed.

Þ Answer:

We added the description in P14, L353-355 as follows:

“The differential expression markers (DEMs) were identified based on the criterion suggested by Béres NJ et al. [39] and Nassirpour R et al. [40], where a fold change greater than 1.5 was used as the threshold.”

  1. The ordinate of Fig. 5 should use different titles. Other figures with the same issue should also be modified accordingly.

Þ Answer:

We appreciate your precise readings. We have changed the ordinate of Figure 5 and Figure 8.

  1. line 151: Due to the many targets of miR-155-5p, it does not seem to indicate that miR-155-5p targets C/EBP β by Fig. 4andFig. 5. At most, there is a connection between miR-155-5p and C/EBP β.

Þ Answer:

We have changed the sentence in P7 L162-164 as follows:

“C/EBPβ is a known target gene of miR-155-5p [19], leading us to postulate that miR-155-5p is connected to the decrease of C/EBPβ under the influence of OTA.”

  1. Fig.7 c: The statistical analysis of C/EBP β does not match the result of the strip.

Þ Answer:

We appreciate your precise readings. We redid the statistical analysis and made the correction of Figure 7c in P10.

  1. The resolution of the ordinateof Fig. 8 is too low to allow for clear viewing.

Þ Answer:

We have made the change in Figure 8.

Finally, we really appreciate the reviews’ critical and valuable comments which have been guidance in revising our manuscript. We believe that our manuscript has been much improved due to the revision based on the reviewer’s suggestions.

Reviewer 2 Report

The manuscript is a further publication of results from a study in which Ochratoxin A effects were investigated in mice given 1 and 3 mg/kg bw 5days per week for 12 weeks. After effects on kidney and liver this manuscript describes effects on the colon of the animals. The number of mice per treatment group was 6.

The following points are causing major problems:

1. the doses of 1 and 3 mg/kg bw per day are deserve a justification because they are 6 orders of magnitude higher than the mean exposure towards Ochratoxin of the European population (0.6 to 17.8 ng/kg bw per day) and even 5 orders  of magnitude than the 95th percentile (2.4 to 51.7 ng/kg bw per day). The results are therefore considered as not relevant for human risk.

2. The concentration of the in vitro study are 25 and 100 nM, equal to 10  and 40  µg/L  whereas the blood concentration of 2000 Swedish young people  (Warensjö Lemming et al., 2015) was  0.028 µg/L (mean) and 0.037 µg/L (average), maximum 0.136 µg/L, which is a factor 1000 lower. Thus, as a justification is not given, the results have to be considered aas not being relevant..

3. In the protocol, described in the reference (Yang, S.A.; Rhee, K.H.; Yoo, H.J.; Pyo, M.C.; Lee, K.W. Ochratoxin A induces endoplasmic reticulum stress and fibrosis in the 405 kidney via the HIF-1alpha/miR-155-5p link. Toxicol Rep 2023, 10, 133-145, to which reference is made, the number of animal per group is given as  N= 6; in the results the number is given as N=3. No explanation is given for this high attrition rate of 50%. Therefore, it is not alowd to draw any conclusion from the results.

4. The discussion does not mention the severe limitations and does not give justifications or explanations. The last sentence  “These findings shed new light on the involvement of miR-155-5p in intestinal fibrosis and highlight its potential as a therapeutic target for addressing intestinal fibrosis and related conditions” is absolutely not justified by the findigns.

In summary, because of the severe limitations the manuscript cannot be considered to be acceptable for publication.

Author Response

Reviewer 2

Comments and Suggestions for Authors

The manuscript is a further publication of results from a study in which Ochratoxin A effects were investigated in mice given 1 and 3 mg/kg bw 5days per week for 12 weeks. After effects on kidney and liver this manuscript describes effects on the colon of the animals. The number of mice per treatment group was 6.

The following points are causing major problems:

  1. the doses of 1 and 3 mg/kg bw per day are deserve a justification because they are 6 orders of magnitude higher than the mean exposure towards Ochratoxin of the European population (0.6 to 17.8 ng/kg bw per day) and even 5 orders  of magnitude than the 95thpercentile (2.4 to 51.7 ng/kg bw per day). The results are therefore considered as not relevant for human risk.

Þ Answer:

First of all, we’d like to express our gratitude to the reviewer for the careful and critical reading of our manuscript. With highlighted in green color text, we made the correction in response.

We greatly value your concern, and we have endeavored to incorporate the most suitable response possible in page (P)12, line (L) 241-248 as follows:

“It should be emphasized that the doses of OTA used in the current animal study, as well as in numerous previous studies referenced in the literature [28,29], were orders of magnitude higher than the average exposure of the human population to OTA [30]. The selection of doses for this study was based on the recent mouse studies reviewed by the European Food Safety Authority [30]. The selection of the lower dose (1 mg/kg body weight (B.W.)) was based on the lowest observed adverse effect level (LOAEL) identified in a 45-day oral study, where a dose of 1.5 mg/kg B.W. elicited a response in kidney antioxidants. Our experiments also included a higher dose (3 mg/kg B.W.).”

  1. The concentration of the in vitro study are 25 and 100 nM, equal to 10  and 40  µg/L whereas the blood concentration of 2000 Swedish young people  (Warensjö Lemming et al., 2015) was  0.028 µg/L (mean) and 0.037 µg/L (average), maximum 0.136 µg/L, which is a factor 1000 lower. Thus, as a justification is not given, the results have to be considered as not being relevant..

Þ Answer:

Our objective was to study the outcome of OTA intoxication. Our previous study reported renal and liver toxicity with 200 nM of OTA (Pyo MC et al, 2020; Shin HS et al, 2020). In other studies of OTA treatment on Caco-2 cells, the concentrations were 0.2 μM and 20 μM (Gao Y et al., 2018). In this study, we have found that 100 nM of OTA induced intestinal fibrosis.

  1. In the protocol, described in the reference (Yang, S.A.; Rhee, K.H.; Yoo, H.J.; Pyo, M.C.; Lee, K.W. Ochratoxin A induces endoplasmic reticulum stress and fibrosis in the 405 kidney via the HIF-1alpha/miR-155-5p link. Toxicol Rep 202310, 133-145, to which reference is made, the number of animal per group is given as  N= 6; in the results the number is given as N=3. No explanation is given for this high attrition rate of 50%. Therefore, it is not allowed to draw any conclusion from the results.

Þ Answer:

All in vivo experiments had the sample number of 6, excluding the histopathological data in Figure 1 and NGS data displayed in Figure 4. This was due to limited colon tissues available as colon tissues of a single mouse was subjected to mRNA and protein isolation, along with NGS analysis and histopathological analysis. Therefore, all 6 colon tissues per group were used in real-time qPCR and western blot, while 3 of them were used for NGS analysis and the other 3 for histopathological analysis.

  1. The discussion does not mention the severe limitations and does not give justifications or explanations. The last sentence  “These findings shed new light on the involvement of miR-155-5p in intestinal fibrosis and highlight its potential as a therapeutic target for addressing intestinal fibrosis and related conditions” is absolutely not justified by the findigns.

Þ Answer:

The discussions were modified, and we changed the last sentence in P13 L313-317 as follows:

“These findings may contribute to our understanding of the involvement of miR-155-5p in intestinal fibrosis and suggest its potential as a therapeutic target for addressing intestinal fibrosis and related conditions. Nevertheless, further studies are necessary to completely understand the function and precise interaction of TGF-β, miR-155-5p and C/EBPβ.”

  1. In summary, because of the severe limitations the manuscript cannot be considered to be acceptable for publication.

Þ Answer:

We appreciate your valuable comments to improve our manuscript. Please evaluate our responses to your comments. We respectfully request that you reconsider your decision.

Finally, we really appreciate the reviews’ critical and valuable comments which have been guidance in revising our manuscript. We believe that our manuscript has been much improved due to the revision based on the reviewer’s suggestions.

Reviewer 3 Report

This article, in my opinion, presents the following issues.

The rationale for the experiments (the relation between OTA, miRNA and pathological outcomes) it is not clearly stated or supported by previous findings

Title should clearly indicate that results are obtained in mice and in vitro.

The design of the experiments on both mice and cultured cells is unclear (how and how much mices/tissues were assigned to experimental groups, and so on); some important details should be stated: it is not sufficient to refer to another study (moreover, in line 290, the reference to another article by the same authors reveals their names that should have been unavailable).

The measure units seems lacking or unclear.

The summary statistics related to statistical analyses should be given (in a table).

What is the theoretical justification of statistical tests employed or why a different statistical test was applied to "in vitro" and "in vivo" results?

In all data plots within the figures the standard deviation is shown only above the mean (it should also been shown below the mean).

Line 137: which test does yield such a p value? what are the "cut-off" values?

Lines 2,3,6,15,16 and in other sections : abbreviations should be explained previously.

Line 12: "highest elevated" is not a good english expression.

Line 73: concentration amount should be stated.

Lines 137-138: it is unclear the meaning of "Validation...were performed by real-time q-PCR" (a similar expression is also present in lines 170 and 186).

Figure 4: plots contained in this figure should better explained.

In conclusion, the scientific/methodological setting seems weak: the  rationale is not clear, design of experiment lacking of information, the presentation of quantitative results seems inappropriate and other issues indicated above.

So, my suggestion to Authors is to rewrite the article keeping into account the above considerations.

No comments

Author Response

Reviewer 3

Comments and Suggestions for Authors

This article, in my opinion, presents the following issues.

  1. The rationale for the experiments (the relation between OTA, miRNA and pathological outcomes) it is not clearly stated or supported by previous findings

Þ Answer:

First of all, we’d like to express our gratitude to the reviewer for the careful and critical reading of our manuscript. With highlighted in green color text, we made the correction in response.

We have added explanations to outline the relationship between OTA, miRNA and fibrosis progression in page (P)12-13, line (L) 272-295 as follows:

“In our study, we observed an increase in miR-155-5p expression in both mice colon tissues and Ca-co-2 cells when exposed to OTA. Interestingly, the inhibition of miR-155-5p resulted in decreased expressions of FN and α-SMA, while the expression of E-cad was increased. These findings lead us to infer that miR-155-5p is likely involved in the progression of fibrosis in the intestine as well. A comprehensive assessment of the signaling pathway underlying the interaction between miR-155-5p and fibrosis development is required. 

MiR-155-5p has been identified to target C/EBPβ, and inhibiting C/EBPβ through miR-155-5p can lead to EMT, invasion, and metastasis in breast cancer. [19]. C/EBPβ is known to be a factor in TGF-β regulated signaling pathway, including fibrotic progressions [32]. In accordance with the study in breast cancer, our studies revealed that C/EBPβ had decreased under OTA treatment, while the expression of miR-155-5p had increased. C/EBPβ involvement in fibrotic development is somewhat divisive, since both elevation and downregulation of C/EBPβ protein levels have been observed in hepatic and pulmonary fibrosis [33]. In our investigation, however, miR-155-5p inhibition restored C/EBP expression. suggesting that OTA treatment promoted miR-155-5p expression, leading to C/EBPβ loss. Furthermore, TGF-β inhibition increased C/EBPβ expression, which was blocked by miR-155-5p expression. Interestingly, the inhibition of miR-155-5p did not affect TGF-β expression, suggesting that TGF-β might be responsible to activate miR-155-5p. These findings suggest that OTA treatment of the colonic intestine induced TGF-β expression, which was then followed by an increase in miR-155-5p and a reduction of C/EBPβ. Nevertheless, further studies should uncover and validate the exact mechanism of the regulation of fibrotic progression by miR-155-5p and C/EBPβ in TGF-β mediated pathway.”

  1. Title should clearly indicate that results are obtained in mice and in vitro.

Þ Answer:

We changed the title as follows:

“MiR-155-5p elevated by Ochratoxin A induces intestinal fibrosis and epithelial-to-mesenchymal transition through TGF-β regulated signaling pathway in vitro and in vivo

  1. The design of the experiments on both mice and cultured cells is unclear (how and how much mices/tissues were assigned to experimental groups, and so on); some important details should be stated: it is not sufficient to refer to another study (moreover, in line 290, the reference to another article by the same authors reveals their names that should have been unavailable).

Þ Answer:

We added the details of both mice and cell experiment designs in P14 L334-338 as follows:

“The design and procedure of the animal study conducted in this paper were described in detail in the previous report [21]. The mice were divided into 3 groups (n=6 per groups): (1) CON group: 0 mg/kg B.W. (body weight) of OTA, (2) OL group: 1 mg/kg B.W. of OTA, and (3) OH group: 3 mg/kg B.W. of OTA. After OTA treatment for 12 weeks, the colon tissues of 6 mice per group were obtained for subsequent study.”

  1. The measure units seems lacking or unclear.

Þ Answer:

The article was put through thorough checking and clarified the units in Figure 1.

  1. The summary statistics related to statistical analyses should be given (in a table).

Þ Answer:

We have stated the statistical analysis method at 4.10 Statistical analysis in P15 L414-417: line 388 as follows:

“A one-way ANOVA method followed by Duncan’s multiple range test (in vivo) and Tukey’s multiple range test (in vitro) were performed. Different letters above the bars rep-resent significant differences at p value lower than 0.05 using SAS version 9.4 (SAS Insti-tute, Cary, NC, USA).”

  1. What is the theoretical justification of statistical tests employed or why a different statistical test was applied to "in vitro" and "in vivo" results?

Þ Answer:

We applied Duncan’s multiple range test to test significant differences in in vivo experiments because the in vivo experiments have the number of samples at 6. This is due to the procedure being based on the comparison of the range of a subset of the sample means with a calculated least significant range; this least significant range increases with the number of samples means in the subset (Bewick V et al., 2004). Since the results of the in vitro experiments of this study is done with triplicate samples, we opted for Tukey’s standardized ranged test for comparison.

  1. In all data plots within the figures the standard deviation is shown only above the mean (it should also been shown below the mean).

Þ Answer:

The changes were made throughout the Figures.

  1. Line 137: which test does yield such a p value? what are the "cut-off" values?

Þ Answer:

We added the description in P5 L136-138 as follows:

“OTA exposure has induced 42 DEMs in the colon tissues of mice (p < 0.05 and fold change > 1.5).”

The sentence was rewritten for clearness in P5 L136-138 as follows:

“This phenomenon was validated in mice colon tissues and Caco-2 cells, where miR-155-5p was significantly (p<0.05) upregulated under OTA treatment (Figure 4c and 4d).”

  1. Lines 2,3,6,15,16 and in other sections: abbreviations should be explained previously.

Þ Answer:

The abbreviations were explained in Pl L9-10, 12, 17, 19-20 (the abstract) and P4 L101-102 (other section).

  1. Line 12: "highest elevated" is not a good English expression.

Þ Answer:

We replaced the word in P1 L16 as follows:

“Following OTA treatment, miR-155-5p was the most elevated miRNA by next-generation sequencing.”

  1. Line 73: concentration amount should be stated.

Þ Answer:

We added the information in P2 L74-75 as follows:

“We conducted a study on the changes in miRNA expression in the colon following OTA exposure, as well as the development of intestinal fibrosis influenced by 1 and 3 mg/kg B.W. of OTA in mice colon tissues, as well as 25 nM and 100 nM of OTA on Caco-2 cells.”

  1. Lines 137-138: it is unclear the meaning of "Validation...were performed by real-time q-PCR" (a similar expression is also present in lines 170 and 186).

Þ Answer:

The sentences in P7 L148-150, P8 166-168, and P9 L184-186 were all modified to provide clearer meaning:

“Validation of miRNA NGS analysis of miR-155-5p in (c) mice colon tissues and (d) Caco-2 cells were performed by measuring miR-155-5p expressions using real-time qPCR”

“Identification of C/EBPβ as miR-155-5p target. mRNA expression of C/EBPβ (a) in vivo and (b) in vitro was evaluated. Protein expression of C/EBPβ (c) in vivo and (d) in vitro was evaluated”

“(a) The expressions of miR-155-5p were measured, and(b) mRNA and (c) protein expressions of fibrosis and EMT markers were examined”

  1. Figure 4: plots contained in this figure should better explained.

Þ Answer:

The explanation was further provided in the figure description in Figure 4.

  1. In conclusion, the scientific/methodological setting seems weak: the rationale is not clear, design of experiment lacking of information, the presentation of quantitative results seems inappropriate and other issues indicated above.

Þ Answer:

We appreciate your valuable comments to improve our manuscript. Please evaluate our responses to your comments.

Finally, we really appreciate the reviews’ critical and valuable comments which have been guidance in revising our manuscript. We believe that our manuscript has been much improved due to the revision based on the reviewer’s suggestions.

Round 2

Reviewer 1 Report

Fig.6b, Fig.6c, etc: It is recommended to label statistical identifiers in alphabetical order. The statistical identifier “ac” of C/EBPβ in Fig.6b is clearly incorrect. And check all figures.

Minor editing of English language required

Author Response

Reviewer 1

Comments and Suggestions for Authors

Fig.6b, Fig.6c, etc: It is recommended to label statistical identifiers in alphabetical order. The statistical identifier “ac” of C/EBPβ in Fig.6b is clearly incorrect. And check all figures.

--> Answer:

First of all, we’d like to express our gratitude to the reviewer for the careful and critical reading of our manuscript. We made the correction in response.

We have made corrections of statistical identifiers. Please see pages 9-10, Figures 6b, 6c, 7b, and 7c.

Comments on the Quality of English Language

Minor editing of English language required

--> Answer:

"We carefully made minor edits throughout the text to improve the clarity and quality of the English language."

Finally, we really appreciate the reviews’ critical and valuable comments which have been guidance in revising our manuscript. We believe that our manuscript has been much improved due to the revision based on the reviewer’s suggestions.

Reviewer 2 Report

Thank you for taking over some points of criticism in your revised version of the manuscript. However, some of the points were addressed only in the response lacking a revision of the text in the revised manuscript.

Concerning point No. 2

  1. The concentration of the in vitro study are 25 and 100 nM, equal to 10  and 40  µg/L whereas the blood concentration of 2000 Swedish young people  (Warensjö Lemming et al., 2015) was  0.028 µg/L (mean) and 0.037 µg/L (average), maximum 0.136 µg/L, which is a factor 1000 lower. Thus, as a justification is not given, the results have to be considered as not being relevant.

and the answer

Þ Answer:

Our objective was to study the outcome of OTA intoxication. Our previous study reported renal and liver toxicity with 200 nM of OTA (Pyo MC et al, 2020; Shin HS et al, 2020). In other studies of OTA treatment on Caco-2 cells, the concentrations were 0.2 μM and 20 μM (Gao Y et al., 2018). In this study, we have found that 100 nM of OTA induced intestinal fibrosis.

I miss a sentence saying that the in vitro concentrations are 1000 fold lower than the blood concentration in healthy subjects (Warensjö Lemming et al., 2015) and they reflect only concentrations following high intake of OTA causing intoxications (please, add a reference).

Concerning point No. 3

  1. In the protocol, described in the reference (Yang, S.A.; Rhee, K.H.; Yoo, H.J.; Pyo, M.C.; Lee, K.W. Ochratoxin A induces endoplasmic reticulum stress and fibrosis in the 405 kidney via the HIF-1alpha/miR-155-5p link. Toxicol Rep 202310, 133-145, to which reference is made, the number of animal per group is given as  N= 6; in the results the number is given as N=3. No explanation is given for this high attrition rate of 50%. Therefore, it is not allowed to draw any conclusion from the results.

Þ Answer:

All in vivo experiments had the sample number of 6, excluding the histopathological data in Figure 1 and NGS data displayed in Figure 4. This was due to limited colon tissues available as colon tissues of a single mouse was subjected to mRNA and protein isolation, along with NGS analysis and histopathological analysis. Therefore, all 6 colon tissues per group were used in real-time qPCR and western blot, while 3 of them were used for NGS analysis and the other 3 for histopathological analysis.

I miss a sentence explaining what you explain in your answer. It would also be helpful to know whether the selection of the tissues taken for NGS analysis or for histopathological analysis was randomized.

Author Response

Reviewer 2

Comments and Suggestions for Authors

Thank you for taking over some points of criticism in your revised version of the manuscript. However, some of the points were addressed only in the response lacking a revision of the text in the revised manuscript.

Concerning point No. 2

  1. The concentration of the in vitro study are 25 and 100 nM, equal to 10  and 40  µg/L whereas the blood concentration of 2000 Swedish young people  (Warensjö Lemming et al., 2015) was  0.028 µg/L (mean) and 0.037 µg/L (average), maximum 0.136 µg/L, which is a factor 1000 lower. Thus, as a justification is not given, the results have to be considered as not being relevant.

and the answer

Þ Answer:

Our objective was to study the outcome of OTA intoxication. Our previous study reported renal and liver toxicity with 200 nM of OTA (Pyo MC et al, 2020; Shin HS et al, 2020). In other studies of OTA treatment on Caco-2 cells, the concentrations were 0.2 μM and 20 μM (Gao Y et al., 2018). In this study, we have found that 100 nM of OTA induced intestinal fibrosis.

I miss a sentence saying that the in vitro concentrations are 1000 fold lower than the blood concentration in healthy subjects (Warensjö Lemming et al., 2015) and they reflect only concentrations following high intake of OTA causing intoxications (please, add a reference).

==> Answer:

First of all, we’d like to express our gratitude to the reviewer for the careful and critical reading of our manuscript. With highlighted in green color text, we made the correction in response.

We appreciate your suggestions and added the explanation in page 12, line 248-257 as follows:

“Our in vitro experiments utilized OTA concentrations as low as 25 nM, which were determined based on previous research. Renal and liver toxicity was documented by Pyo MC et al. [31] and Shin HS et al. [8] at an OTA concentration of 200 nM. Additionally, Gao Y et al. [32] employed concentrations of 0.2 μM and 20 μM in their study involving OTA treatment on Caco-2 cells. However, it is important to note that that our selected concentrations only represent levels associated with high OTA intake leading to intoxications [6, 28, 29], and they are approximately 1000 times lower than the blood concentration observed in healthy individuals. A study involving 2000 Swedish adults reported a maximum blood concentration of OTA at 0.136 μg/L [33].”

Concerning point No. 3

  1. In the protocol, described in the reference (Yang, S.A.; Rhee, K.H.; Yoo, H.J.; Pyo, M.C.; Lee, K.W. Ochratoxin A induces endoplasmic reticulum stress and fibrosis in the 405 kidney via the HIF-1alpha/miR-155-5p link. Toxicol Rep 202310, 133-145, to which reference is made, the number of animal per group is given as  N= 6; in the results the number is given as N=3. No explanation is given for this high attrition rate of 50%. Therefore, it is not allowed to draw any conclusion from the results.

Þ Answer:

All in vivo experiments had the sample number of 6, excluding the histopathological data in Figure 1 and NGS data displayed in Figure 4. This was due to limited colon tissues available as colon tissues of a single mouse was subjected to mRNA and protein isolation, along with NGS analysis and histopathological analysis. Therefore, all 6 colon tissues per group were used in real-time qPCR and western blot, while 3 of them were used for NGS analysis and the other 3 for histopathological analysis.

I miss a sentence explaining what you explain in your answer. It would also be helpful to know whether the selection of the tissues taken for NGS analysis or for histopathological analysis was randomized.

==> Answer:

We appreciate your suggestions and stated the explanation in page 14, line 356-357 and 360-361 as follows:

“3 mice colon tissues were chosen randomly from each group for this study (n=3).”

“Three mice colon tissues from each group were randomly selected and subjected to miRNA-sequencing.”

Finally, we really appreciate the reviews’ critical and valuable comments which have been guidance in revising our manuscript. We believe that our manuscript has been much improved due to the revision based on the reviewer’s suggestions.
